# Using Genetic Programming to Identify Characteristics of Brazilian Regions in Relation to Rural Credit Allocation

Adolfo Vicente Araújo [1,*], Caroline Mota [1] and Sajid Siraj [2]

1  Department of Industrial Engineering, Federal University of Pernambuco, Recife 50670-901, Brazil
2  Centre for Decision Research, Leeds University Business School, University of Leeds, Leeds LS2 9JT, UK
*  Correspondence: adolfo.vicente@ufpe.br

**Abstract:** Rural credit policies have a strong impact on food production and food security. The attribution of credit policies to agricultural production is one of the main problems preventing the guarantee of agricultural expansion. In this work, we conduct family typology analysis applied to a set of research data to characterize different regions. Through genetic programming, a model was developed using user-defined terms to identify the importance and priority of each criterion used for each region. Access to credit results in economic growth and provides greater income for family farmers, as observed by the results obtained in the model for the Sul region. The Nordeste region indicates that the cost criterion is relevant, and according to previous studies, the Nordeste region has the highest number of family farming households and is also the region with the lowest economic growth. An important aspect discovered by this research is that the allocation of rural credit is not ideal. Another important aspect of the research is the challenge of capturing the degree of diversity across different regions, and the typology is limited in its ability to accurately represent all variations. Therefore, it was possible to characterize how credit is distributed across the country and the main factors that can influence access to credit.

**Keywords:** rural credit; criteria analysis; family farming; genetic programming; machine learning

## 1. Introduction

Food insecurity is a global challenge, and the Central Rural Work Conference in 2021 pointed out that in order to ensure food security, attention should be paid to adjusting rural credit [1]. The allocation of credit to agricultural production is one of the main problems in ensuring agricultural expansion, since the credit is financed with lower interest rates than those adopted in the market [2]. Rural credit consists of loans provided by financial institutions to producers and rural cooperatives [3]. According to data from the Central Bank of Brazil [4], in 1999, the rural credit/agricultural GDP ratio was approximately 24%, and in 2018, it reached approximately 61%. The increase in rural credit and its importance to Brazilian agricultural policy become even more relevant when we consider studies that show the impact of positive rural credit on agricultural variables such as production value, agricultural products, agribusiness products, and total factor productivity [5,6].

Many studies from different parts of the world have shown significant economic and social elevation for the beneficiaries of rural credit programs [7]. In addition, many specialists believe that large farmers linked to global companies are key actors, based on both the size of their production and the potential for agricultural intensification. However, for other specialists, diversified systems of family agriculture allow for greater potential for development. Other experts analyze the impact of public policies on the well-being and economic improvement of rural areas around the world, concluding that rural credit policies represent an essential factor in the development of agriculture [8,9]. Although some studies have addressed rural credit, no study has sought to understand how rural credit is allocated to farmers, and no specific technique or methodology has been used to properly assess rural credit for the actors involved.

The difficulty in finding studies related to the conception of rural credit is due to the complexity of the rural credit system. For example, the government of Brazil has a complex set of funding sources and rural credit programs and offers a range of credit lines targeted at producers with different income levels and property sizes. However, flaws in the current design lead to substantially reduced access and distribution of rural credit. This diminishes the benefits of the policy and threatens the country's ability to balance its agricultural productivity with environmental preservation [3].

During the 2017 agricultural year, the Brazilian rural credit system offered 18 sources of financing. Important differences existed among these funding sources in terms of beneficiary qualification criteria and funding conditions. This generated a complex system and made it difficult for creditors and debtors to define the best credit option in each case. The complex set of rules for the different credit lines was not only excessively complicated for the agents operating the system but also generated distortions in the allocation of resources [10].

It is argued that the current structure favors the three public banks: (1) Banco do Brasil, the main creditor in the sul, sudeste, and centro-oeste revisions; (2) Banco do Nordeste, the main creditor in the nordeste region; and (3) Banco da Amazônia, the main creditor in the norte region. Considering the BRL 218 billion made available during the 2017–2018 harvest, which represents approximately 7% of the balance of credit operations in the entire financial system, rural credit is restricted to a small portion of rural establishments in Brazil. According to the Brazilian Institute of Geography and Statistics (IBGE) [11] only 15.5% of rural establishments had access to the rural credit system. Among those who did not obtain it, 42.8% stated reasons that did not prevent them from taking out rural credit. Such values correspond to the existing paradigm at specialized rural credit institutions [12] and, consequently, to the hypothesis of rural credit rationing in the country [13,14]. Banco do Brasil, the public bank, provided 47% of all loans in the 2017 fiscal year. Itaú and Bradesco, two of the most important private banks in Brazil, together with Banco do Brasil, provided about 60% of all loans in the same period. Credit cooperatives were responsible for another 15% of rural credit in 2017 [10].

In this respect, guaranteeing access to rural credit for all farmers, and not just for a minority, is considered a vital requirement for sustainable economic growth and for improving the quality of life in less developed countries. In order to understand the rural credit allocation process, we used family typology analysis applied to a research dataset to characterize different regions. Previous studies used several approaches to determine rural credit but did not take into account the complexity of heterogeneous family farming systems. There is no work in the literature that characterizes how rural credit, an important public policy that has been adopted by several countries, is conceived. Therefore, it was necessary to investigate and characterize the allocation of rural credit to different Brazilian regions using an artificial intelligence technique. The technique used in this study was designed using user-defined terms to identify the importance and priority of each criterion used for each region. Thus, farmers who are interested in accessing rural credit may have a better decision-making capacity in relation to the most important criteria in a given region.

This paper is organized as follows: Section 2 presents the bibliographic review, and Section 3 describes the genetic programing. In Section 4, we discuss the method, and in Section 5, we discuss the results. Finally, we provide the conclusion in Section 6.

## 2. Bibliographic Review

The period of 1995 to 2022 was selected for this systematic review, because the National Program to Strengthen Family Farming (PRONAF) was created in 1995. PRONAF was introduced to support family farmers because there was an increase in farmer indebtedness and non-payment [15].

For the systematic review, first, the criteria for inclusion were defined, followed by the systematic implementation of defined search strings across the Web of Science, Scopus, and SciELO databases. To achieve quality and ensure relevance to the topic, only articles

that were published in respected international peer-reviewed journals were shortlisted. All other publications, including conference articles, post-graduation theses, and editorial notes, were excluded from this review. This approach allowed for a description of the techniques, methodologies, applied tools, and possible trends in the bibliography, as well as the gaps in research on rural credit. Peer-reviewed articles were included in the analysis based on the criteria listed in Table 1.

**Table 1.** Inclusion criteria for the systematic review.

| Category | Inclusion Details |
|---|---|
| Language | Non-English studies have been excluded |
| Publication date | Any studies prior to 1995 have been excluded |
| Peer-reviewed | Only peer-reviewed articles have been included |
| Geography | Only studies related to rural credit around the world |
| Primary data | Only the studies presenting primary data |
| Subjects | Only those studies that relate to family farming |
| Treatment | The studies must be related to rural credit, or to one of its practices or pseudonyms, as a financing policy |
| Rural credit aspect | Studies must be related to one or more aspects of Rural Credit: ● Criteria (quantity) ● Methodology ● Tools ● Distribution |

The next step was to define the keywords (search strings) to be used in the first filter to select the articles. The words defined in advance were "rural credit" and "family farming", "rural credit distribution", "rural credit measurement", and "indicators of rural credit" as terms on one axis of the theoretical background. The keywords connected to rural credit and family farming were combined to form the other axis of the research. Then, the keywords were constructed, as shown in Table 2.

**Table 2.** Keywords.

| Keywords | TOPIC—Search Strings in the Title and Summary of the Articles |
|---|---|
| P1 | Topic = ("Rural Credit") AND Topic = ("Family Farming") |
| P2 | Topic = ("Rural credit distribution") AND Topic = ("Family Farming") |
| P3 | Topic = ("Rural credit measurement") AND Topic = ("Family Farming") |
| P4 | Topic = ("Indicators for rural credit") AND Topic = ("Family Farming") |
| P5 | Topic = ("Model used in rural credit") AND Topic = ("Family Farming") |
| P6 | Topic = ("Distribution of public resources in agriculture") AND Topic P1 |
| P7 | Topic = ("Agriculture performance") AND Topic P1 |
| P8 | Topic = ("Tools used in agriculture") AND Topic P1 |
| P9 | Topic = ("Criteria for public resources") AND Topic P1 |
| P10 | Topic = ("Methodology used in sustainable agriculture") AND Topic P1 |

Then, the articles that were considered to be aligned with the theme of this research were analyzed using the Journal Citation Reports (JCR) of their respective publishers. The objective was to verify whether they possessed a JCR impact factor (IF) score different from zero. As a quality standard, only journals with a JCR IF greater than zero were included in this research.

*Selection of the Theoretical Framework*

Using the inclusion criteria for the systematic review and the defined keywords, the search of the database yielded 2699 results. The summaries of those 2699 articles were then analyzed to check whether their themes were aligned with the research. After this initial analysis, 2597 references were excluded for not being aligned with this research or for being duplicates. Then, the 102 remaining articles were analyzed using the JCR criterion. It was observed that 49 did not meet the requirements, meaning that the JCR indicator was zero. Hence, the remaining 53 articles were aligned with regard to the title and abstract and met

the JCR requirement established by the researcher. Those articles were then read in full in order to evaluate whether they would contribute valuable information to this study. In this process, three articles were excluded, resulting in 50 articles, as shown in Figure 1.

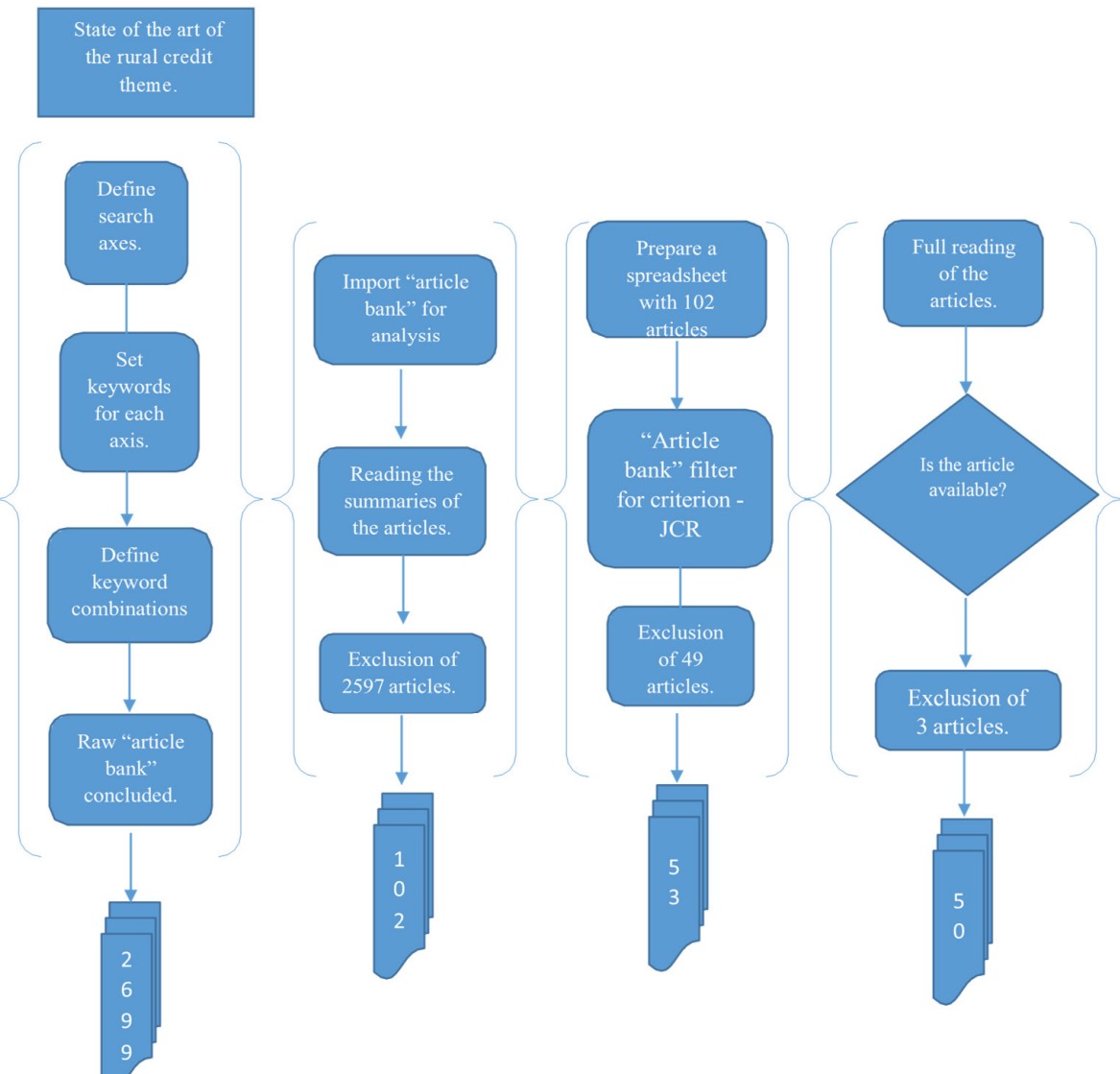

**Figure 1.** Flowchart illustrating the systematic review process.

Figure 2 shows the systematic process for the visualization of techniques and criteria from the identification of 50 papers deemed eligible for full analysis. For this research, we divided the articles into five techniques that were used in their respective work: statistical, computational, multicriteria, exploratory review, and other. The aim was to find a technique that could apply to the work, considering its limitations.

A total of 72% of articles reported a quantitative approach, among which 16% used statistical techniques to solve the problems [16–23]. Computational techniques were used in 28% [8,24–37]; multicriteria techniques were used in 22% [38–48]; literature review, a qualitative approach, was used in 28% [49–62]; and other types were used in 6% [63–65], Table 3.

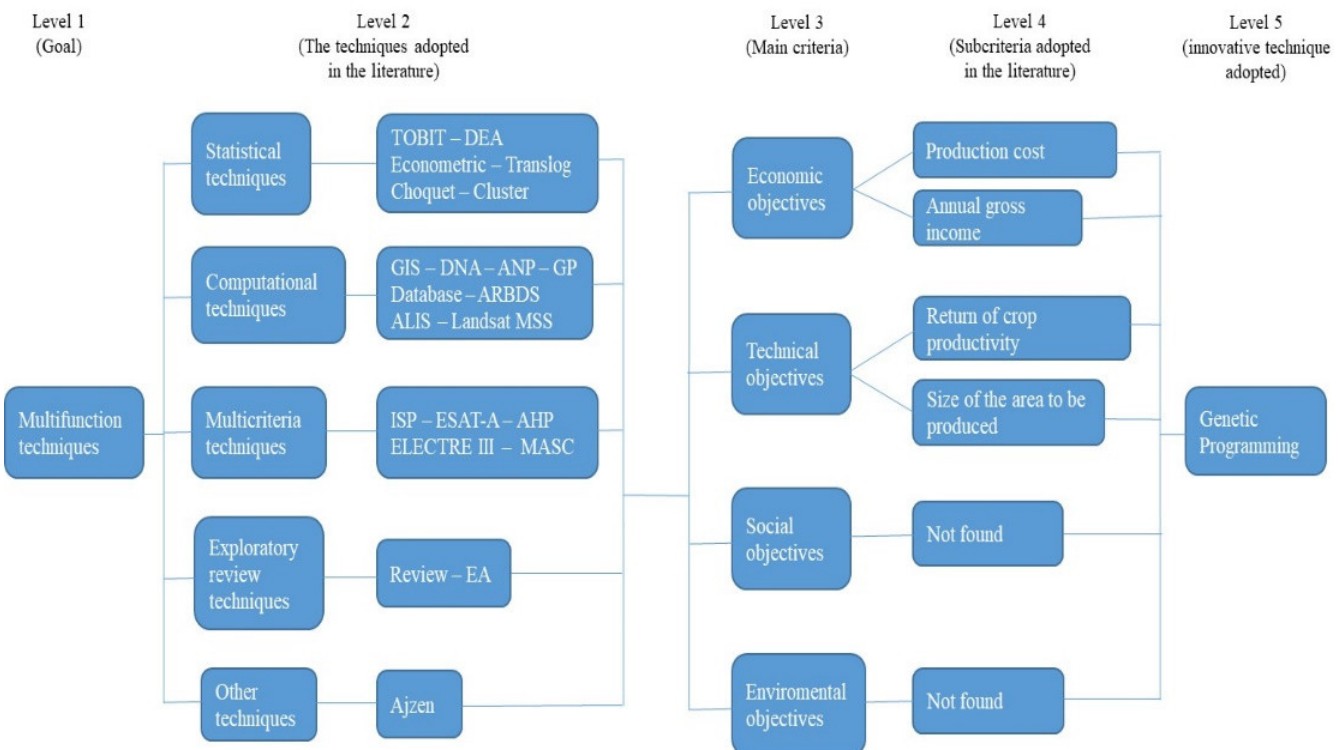

**Figure 2.** Flowchart illustrating the systematic process.

**Table 3.** Percentage of technical approaches.

| Total Percentage of Techniques Adopted in the Literature | |
| --- | --- |
| Statistical techniques | 16% |
| Computational techniques | 28% |
| Multicriteria techniques | 22% |
| Literature review | 28% |
| Other types | 6% |

## 3. Genetic Programming

Genetic programming has attracted attention for solving credit problems, both academically and empirically, and is mainly implemented for problems with less information. For genetic programming, it fits as a machine learning method, but unlike with other learning methods, the answer is already a readable and interpretable model in the form of a program. When we are able to define the language in which the program will be written, the obtained program will be explicitly interpretable and will not require an additional step to understand the decision-making process. For example, in [62], the authors performed an analysis of credit scoring models for public banks using two evaluation criteria: the average correct classification (ACC) rate and estimated misclassification cost (EMC). The results revealed that GP had a higher ACC and lower EMC compared to other techniques [62]. Another example supports the recommendation to use the "genetic programming in two stages" (2SGP) technique to deal with credit scoring problems, incorporating IF–THEN rules and more complex discriminating functions. Based on the numeric results, the conclusion is that 2SGP can offer better precision than other models, and this improvement can result in significant savings [63].

Other studies related to credit supply describe a relationship to economic growth. For example, in [66], the authors used the coherence test to answer the following question: (i) does the offer of bank credit lead to economic growth? Another study [67] examined

the causal link of sectoral economic credit growth in Australia using a coherence test. The results for both studies determined according to dynamic time and frequency analysis reveal that economic growth generates credit for the agricultural sector in the long term.

## 4. Method

A structured questionnaire-based survey was conducted using the IBGE data collection platform to collect pooled information from 264,359 agricultural households. Through multivariate analysis, a typology of family farming types was developed based on fourteen area ranges, totaling seventy groups of agricultural households. Subsequently, discussions of these seventy groups were conducted using criteria developed through the survey. The purpose of the discussions was to validate the typology of family farming and to explore and characterize the rural credit granted to agricultural households. In addition, the genetic programming technique was introduced through the Bitbucket website (https://bitbuc ket.org/ciml (accessed on 1 June 2022)), which is public and allows for free collaboration. The implementation is in C++, and all tests were performed on the Debian/Ubuntu Linux operating system to obtain conclusions that could help in decision making.

### 4.1. Study Area

For this research, the study area was divided into five regions: Norte, Nordeste, Sudeste, Sul, and Centro-Oeste (Figure 3).

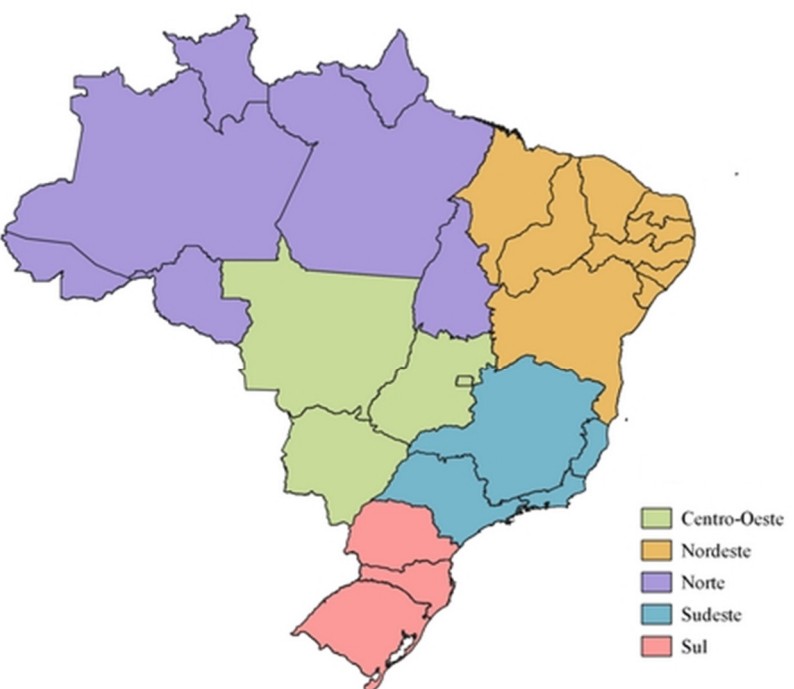

**Figure 3.** Map of Brazil showing the study areas.

### 4.2. Survey

The dataset covers the period from June 2007 to July 2017 [10]. To evaluate the criteria, first, a document is used, which is the gateway to public policies [11,68]. Considering the legal and regulatory framework in force on 30 September 2017, the information available in the 2017 Agricultural Census, the definitions used in the 2006 Agricultural Census, and the opinions of the technical management of the Agricultural Census (GTA) of IBGE, several algorithms were proposed [69]. When their credit applications were processed, all farmers were evaluated according to the following criteria:

- The total gross income of the agricultural household does not exceed BRL 500,000.
- The area of the property does not exceed four fiscal modules, a unit of measurement in hectares, whose value for each municipality is determined by the National Institute of Colonization and Agrarian Reform (INCRA).
- Farmers predominantly use their own family's labor for the activities of the establishment or enterprise.
- Farmers have a PRONAF Aptitude Declaration (DAP).

In addition to these classification criteria, the proposed model was generated by four independent criteria—production cost, annual gross income, return to crop productivity, and size of the area to be produced—and a dependent criterion: number of establishments that received rural credit (Table 4).

**Table 4.** List of variables of the model.

| Variables | Code | Description |
|---|---|---|
| $x_1$ | Territorial Region | Norte, Nordeste, Sudeste, Sul, Centro-Oeste |
| $x_2$ | Typology | Family farming |
| $x_3$ | Area range | Fourteen groups of categorized areas |
| $x_4$ | Productivity | Production value of agricultural establishments (per thousand Reais (BRL)) |
| $x_5$ | Cost | Value of expenses incurred by agricultural establishments (per thousand Reais (BRL)) |
| $x_6$ | Income | Value of income or income obtained by agricultural establishments (per thousand Reais (BRL)) |
| Y | Approved quantity | Quantity of establishments that received rural credit |

With this set of criteria, genetic programming was used to characterize rural credit. For this work, two scenarios were developed in an attempt to find a function that would explain it analytically:

- Scenario 01, using the dataset separately for each region.
- Scenario 02, a generalist function using the entire dataset without separating by region.

Some authors use absolute error for fitness functions [64,70], while others use linear combinations of the mean square error and mean classification error [71]. In this paper, we preferred the latter approach. The fitness function, F, for evolution was calculated as follows:

$$F(ep) = \sum_{i=1}^{n} \frac{|a_i - e_i|}{n}$$

where $F$ is the fitness function, $ep$ is the evolved program, $a_i$ is the actual observation, $e_i$ is the expected (predicted) observation, and $n$ is the sample size.

### 4.3. Typology Construction

The typological approach has previously been used in studies to characterize farming families [72,73]. The advantage of organization and the use of aggregated data is that the data are more stable, and this approach is widely used to build typologies of farmers [74,75]. Therefore, the aggregated data for this research were organized and used for different categories of agricultural areas corresponding to each region (Table 5).

**Table 5.** Sample size determination.

|  | Norte | Nordeste | Sudeste | Sul | Centro-Oeste |
|---|---|---|---|---|---|
| Total population of farmers | 10,764 | 73,612 | 56,229 | 112,830 | 10,924 |
| Sample size (0–0.1 ha) | 38 | 406 | 89 | 38 | 4 |
| Sample size (0.2–0.5 ha) | 49 | 364 | 72 | 27 | 1 |
| Sample size (0.5–1 ha) | 68 | 1455 | 197 | 63 | 1 |
| Sample size (1–2 ha) | 144 | 4069 | 384 | 200 | 7 |
| Sample size (2–3 ha) | 148 | 8641 | 1622 | 585 | 34 |
| Sample size (3–4 ha) | 142 | 6501 | 2584 | 1296 | 65 |
| Sample size (4–5 ha) | 148 | 5506 | 2837 | 1993 | 59 |
| Sample size (5–10 ha) | 486 | 3915 | 3136 | 2809 | 253 |
| Sample size (10–20 ha) | 1023 | 11,837 | 9951 | 17,758 | 964 |
| Sample size (20–50 ha) | 3204 | 11,460 | 13,060 | 32,152 | 1844 |
| Sample size (50–100 ha) | 2621 | 12,536 | 14,788 | 42,114 | 3110 |
| Sample size (100–200 ha) | 2072 | 4783 | 5994 | 13,464 | 2777 |
| Sample size (200–500 ha) | 615 | 1811 | 1403 | 326 | 1378 |
| Sample size (500–1000 ha) | 6 | 328 | 112 | 5 | 427 |

### 4.4. Typology Validation

Evaluating aggregate data involves analyzing and interpreting data that have been combined. The steps used in the research to evaluate the aggregated data, according to [76], are as follows:

- Understand the purpose and context of the data. For this research, we did not want to use the machine learning technique for prediction but rather to characterize rural credit.
- Check the data quality. We used a reliable Brazilian census [11], which is prepared by a trustworthy technical team that carries out mapping in different areas. The aggregated data were certified as complete by eliminating missing data.
- Compare the data. The literature was compared with data obtained from the census to identify similarities and differences.
- Draw a conclusion. Based on the analysis, conclusions and recommendations were made. It is important to be cautious when interpreting data and consider the limitations.

Therefore, evaluating aggregated data requires a systematic approach that involves understanding the purpose and context of the data, checking the data quality, analyzing the data, comparing the data, drawing conclusions, and communicating the results. For this research, the use of aggregate data was justified by the objective of the work, which was to characterize farmers from different regions regarding their willingness to receive rural credit. The interest of the research was to group agricultural households according to total area and not to infer the information of each farmer.

## 5. Results

### 5.1. First Scenario

Following the methods described above, five regions were characterized using the equations generated by the genetic programming technique based on symbolic regression. The five regions presented the output of the model (evolved program). The characterization of regions was carried out according to the variables used in the model (Table 6).

To verify the robustness in terms of whether the characterization generated by the model resembled reality, the model was validated through discussion based on the literature [77]. Equations for each region were developed. From these equations, we identified the variables that allowed for a qualitative comparison (no, low, moderate, or high influence) for the approved number of agricultural establishments in relation to rural credit among regions (Table 7).

**Table 6.** Characterization of regions based on their variables.

| Characterization of Regions Based on Their Variables | Territorial Region | Evolved Program |
| --- | --- | --- |
| $x_3 = area$ <br> $x_6 = income$ | Norte | 3.21507 |
| $x_3 = area$ <br> $x_5 = cost$ | Nordeste | 44.57690 |
| $x_3 = area$ <br> $x_4 = productivity$ <br> $x_6 = income$ | Sudeste | 52.21970 |
| $x_5 = cost$ <br> $x_6 = income$ | Sul | 29.76710 |
| $x_3 = area$ <br> $x_4 = productivity$ <br> $x_5 = cost$ | Centro-Oeste | 0.92433 |

**Table 7.** Characterization of the type of region identified.

| Territorial Region | Equation | Area | Productivity | Cost | Income |
| --- | --- | --- | --- | --- | --- |
| Norte | $ep = \frac{3^{x_3}}{5^{x_3}} x_6$ | Moderate influence | No influence | No influence | High influence |
| Nordeste | $ep = \frac{(x_5)}{(81+(x_3-4))+\left(\frac{1}{x_3}+2\right)^{(x_3-5)}}$ | Moderate influence | No influence | High influence | No influence |
| Sudeste | $ep = \begin{cases} \frac{3^{x_3}}{1027} x_3 \leq 10 \\ \frac{x_6^2}{5x_4} > 10 \end{cases}$ | High influence | No influence | High influence | High influence |
| Sul | $ep = \frac{x_6 - x_5}{93}$ | No influence | No influence | Low influence | High influence |
| Centro-Oeste | $ep = \begin{cases} \frac{x_4}{x_5} x_3 > 5 \\ x_4 x_3 \leq 5 \end{cases}$ | High influence | High influence | Low influence | No influence |

Norte: According to the GP model, the higher the income of farming families, the greater the chance their farms will obtain approved rural credit. Nordeste: According to the GP model, the higher the production cost for farming families, the greater the chance their farms will obtain approved rural credit. This implies that the Nordeste region has high production costs and needs access to rural credit to produce. This reflects reality, as this region has low agricultural production compared to other regions [77]. Sudeste: The GP model allows for inequality, with the characterization of rural credit being represented by the variables area, productivity, and income. According to the model, for farms with an area greater than 10 hectares, income positively influences the allocation of rural credit. When the area is less than or equal to 10 hectares, the function is totally dependent on the area. Sul: According to the GP model, the greater the difference between income and cost for agricultural families, the greater the chance their farms will obtain approved rural credit. Centro-Oeste: The GP model allowed for an inequality, with the characterization of rural credit being represented by the variables area, productivity, and cost. According to the model, for establishments with an area greater than 5 hectares, the ratio of productivity and area has a positive influence on the allocation of rural credit. The cost for this area range has a negative influence. When the area is less than or equal to 5 hectares, the function is represented by the variables area and productivity, and the greater this ratio, the greater the chances that establishments will have approved credit.

### 5.2. Second Scenario

The GP model allowed for an inequality, with the characterization of rural credit being represented by the variables area, productivity, cost, and income. According to the model, for establishments with an area equal to or greater than 20 hectares, the productivity and area ratio positively influences the allocation of rural credit. Cost and income for this area range have a negative influence. When the area is in the range of 10 to 20 hectares, the function is represented by the variables area, productivity, and income, and the greater the positive relationship between the area and productivity, the greater the chances the establishments will obtain approved rural credit. When the area is less than or equal to 10 hectares, cost has a positive influence (Table 8).

**Table 8.** Characterization of Brazil.

| Territory | Equation | Total Area Groups | Evolved Program | Variables |
|---|---|---|---|---|
| Brazil | $ep = \frac{64\left(x_5 - x_3^5 - 3130\right)}{9409}$ | $x_3 \leq 10$ | 646.33000 | $x_3 = area$ <br> $x_5 = cost$ |
| | $ep = \frac{2x_3 + x_4 - x_6}{85}$ | $10 < x_3 < 20$ | | $x_3 = area$ <br> $x_4 = productivity$ <br> $x_6 = income$ |
| | $ep = \frac{(x_3)^3 + (x_3 \times x_4) - x_6}{(x_5)^2}$ | $x_3 \geq 20$ | | $x_3 = area$ <br> $x_4 = productivity$ <br> $x_5 = cost$ <br> $x_6 = income$ |

Different regions have different degrees of need when making decisions based on the analyzed variables. Therefore, according to the characterization of the territory in relation to the allocation of rural credit, this research demonstrates that using a single rule to describe Brazil would not be ideal.

*5.3. Discussion of the Findings*

The advent of models generated via machine learning has impacted the world in different ways, with a presence in practically all areas of knowledge [78]. In some areas, models are applied directly, while in others, they function as a decision-making tool. However, in order to make a decision, it is desirable to understand how it was taken, which is where several learning models fail. A current example is the use of deep learning to generate artificial neural networks: although they are highly accurate, there is a trade-off between accuracy and complexity, making it difficult to extract information about how they work.

The need to explain these models leads to them being restricted or to the use of auxiliary tools capable of generating a more familiar representation of the model. Local Interpretable Model-Agnostic Explanations (LIME) is a tool that can identify an interpretable model that is locally true to the original [78]. Developed by researchers at the University of Washington to achieve greater transparency in terms of what happens inside the model, LIME has become very popular in the community for explaining AI models. When it comes to developing a highly accurate model, explainability is more difficult to achieve due to increasing complexity. For problems with higher dimensions, the lack of explainability is even more evident. Interpretability is an advantage for symbolic regression, as developed by the GP model; in this research, productivity is seen as contributing to the description of "approved rural credit", while the high cost of production is evidenced as "without approval".

For example, studies indicate [66,67] that access to credit provides economic growth, improving agricultural productivity and providing higher incomes for family farmers, as observed by the result of the model for the Sul region. As another example, for the Nordeste region, the cost criterion is relevant, and according to studies [62,77], the Nordeste region has the highest number of family farm households and is also the region with the lowest economic growth. Therefore, it lacks technological development, allowing for higher production costs. Therefore, the decision-making process can be based on a non-black-box model with a clear understanding of how the decision was made (Figure 4).

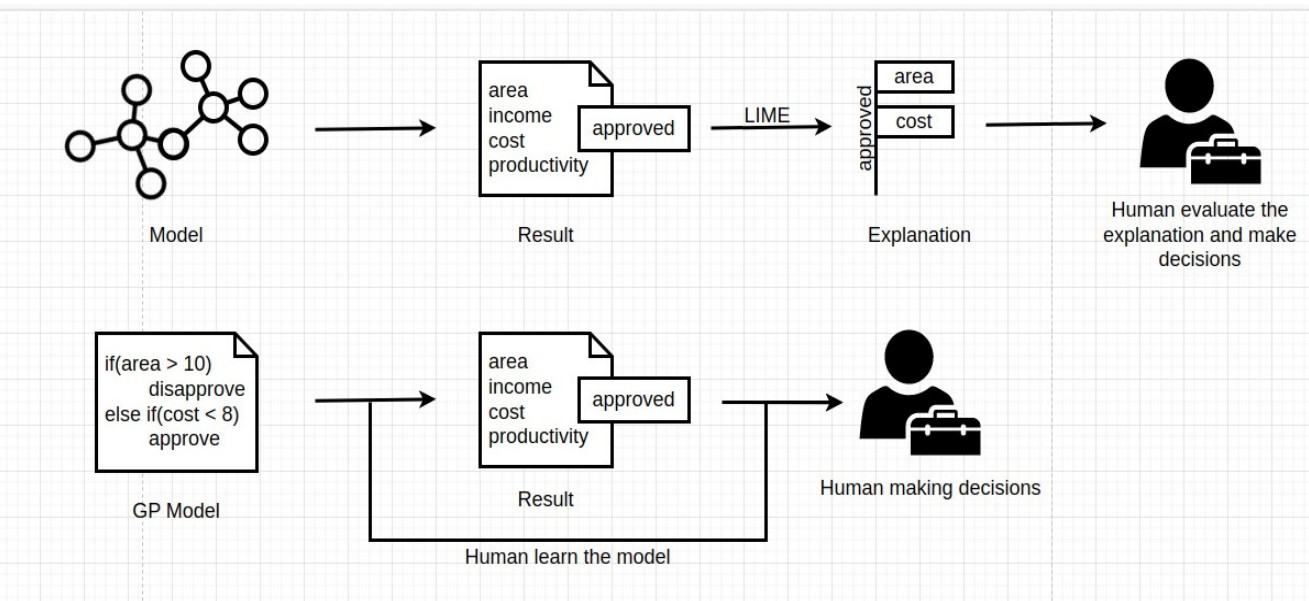

**Figure 4.** A model providing an answer based on data. LIME builds possible explanations according to why that answer was given.

## 6. Conclusions

The improvement of state-sponsored agricultural policies plays a decisive role in maintaining the distribution of rural credit. Groups of family farmers seek to introduce credit instruments to help them become more competitive on the agricultural market. However, the budget constraints of rural credit make it difficult to purchase items such as agricultural inputs. An important aspect discovered by this research is that the allocation of rural credit is not ideal. Another important aspect of this research is the challenge of capturing the degree of diversity in different regions, and the typology is limited in its ability to accurately represent all variations. Given the large territorial extent of Brazil and its characteristic heterogeneity in terms of demographic, economic, and cultural diversity, it is a challenge to develop a rural credit model to be applied universally. However, it is possible to characterize how credit is distributed across the country and the main factors that can influence access to credit; therefore, this is strong and innovative research, since public policies for rural credit are worldwide, and many countries that are in the same condition as Brazil have this problem. Therefore, using the machine learning technique could provide a better understanding of the regions according to the needs of agricultural households. In addition, GP models can help in implementing policies for smallholder support services. The technique used in this study was designed to generate a symbolic response using user-defined terms and identify the importance and priority of each criterion used for each region. Thus, those who are interested in rural credit may have better decision-making ability in relation to the most important criteria in a given territorial group and region.

**Author Contributions:** Conceptualization, A.V.A. and C.M.; Methodology, A.V.A. and S.S.; Validation, A.V.A., C.M. and S.S.; Formal analysis, C.M.; Investigation, A.V.A.; Resources, A.V.A.; Writing–original draft, A.V.A.; Writing–review & editing, A.V.A.; Supervision, C.M. and S.S. All authors have read and agreed to the published version of the manuscript.

**Funding:** This research received no external funding.

**Institutional Review Board Statement:** Not applicable.

**Data Availability Statement:** https://sidra.ibge.gov.br/pesquisa/censo-agropecuario/censo-agropecuario-2017/resultados-definitivos (accessed on 31 March 2023).

**Acknowledgments:** Coordenação de Aperfeiçoamento de Pessoal de Nível Superior (CAPES) and CNPq.

**Conflicts of Interest:** The authors declare no conflict of interest.

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
