# Peer review of "Using Genetic Programming to Identify Characteristics of Brazilian Regions in Relation to Rural Credit Allocation"

_agriculture, doi:10.3390/agriculture13050935_

Round 1

Reviewer 1 Report (Previous Reviewer 1)

Dear Author(s),

The revised version of the submission entitled “A Using genetic programming to identify characteristics of Brazilian regions in relation to rural credit allocation” (Manuscript ID: agriculture-2351589) improved in a very proper manner. The amended manuscript was modified according to the suggested changes and recommendations as formulated throughout the prior review round. Also, proper explanations were provided. Therefore, the manuscript improved significantly.

The English language is adequate.

Author Response

Goodnight.

Thank you for your considerations. I will send the Reviewer's certificate from the journal itself.
Best wishes.

Reviewer 2 Report (New Reviewer)

I suggest clarifying the aims, gaps, and contributions throughout the whole study, especially in the abstract, introduction, and conclusion sections. Furthermore, several studies explained the credit supply that needs to be added in your study such as A) Time-frequency co-movements between bank credit supply and economic growth in an emerging market: Does the bank ownership structure matter?. The North American Journal of Economics and Finance, 2020, 54, 101239. B) Examining the sectoral credit-growth nexus in Australia: a time and frequency dynamic analysis. Econ Comput Econ Cybern Stud Res, 2021, 55(4), 69-84. Third, the policy implications and limitations should be improved. Fourth, the results part should be explained by supporting the prior studies.

It needs some minor modifications.

Author Response

Hello. Thank you for your considerations. Attached is the certificate of review in English by the journal itself. 

Follow the answers:

Reviewer's suggestion

Authors response

1

I suggest clarifying the aims, gaps, and contributions throughout the whole study, especially in the abstract, introduction, and conclusion sections.

We have further clarified the aims, gaps and contributions in the abstract, introduction and conclusion sections.

2

Several studies explained the credit supply that needs to be added in your study:

·  Time-frequency co-movements between bank credit supply and economic growth in an emerging market: Does the bank ownership structure matter? The North American Journal of Economics and Finance, 2020, 54, 101239

·  Examining the sectoral credit-growth nexus in Australia: a time and frequency dynamic analysis. Econ Comput Econ Cybern Stud Res, 2021, 55(4), 69-84.

Thank you for your suggestions. We have cited these studies related to credit supply.

3

Third, the policy implications and limitations should be improved.

We have revised our conclusion section with policy implication and limitations

4

Fourth, the results part should be explained by supporting the prior studies.

We have revised our discussion of results by mentioning prior studies.

Round 2

Reviewer 2 Report (New Reviewer)

The authors considered my comments and the revised version significantly improved.  

It's good.

This manuscript is a resubmission of an earlier submission. The following is a list of the peer review reports and author responses from that submission.

Round 1

Reviewer 1 Report

Dear Esteemed Author(s),

Please find attached the Review Report.

Reviewer 2 Report

Dear authors,

It is my pleasure to read your submitted paper. I have the following comments, which I believe these are crucial elements in justifying the acceptance of the paper to be published.

First, the motivation of the paper and the objective of the paper are rather disconnected. It is difficult to understand why the authors specifically examining the “mathematical methods”, and then how does different mathematical approaches relate to the credit approval? If there is an obvious connection, please motivate this connection in the introduction section. This is also related to the content of the study, the authors were motivated to study the agricultural credit offered by the private sectors or the public sector, such as by government.  

Second, there is very limited information about why using GP is an appropriate method for this study. As far as I read this work, it is rather a meta-analysis or a review of the existing studies. It was poorly explained what is the connection between the section 2.1: Bibliographic review and the section 2.2: Data collection. What does the authors mean “There were 369,927 establishments that obtained financing via rural credit, totaling 192.70 billion reais (BANCO CENTRAL DO BRASIL, 2021).” Is that households received some kind of credits? Or are these some existing publication in three data bases you have searched. Table 4 was also misleading. It is unclear what does the variable x1, x2, till x6 measured?

Third, the results section shows limited connection with the methods and materials that the authors have proposed. Figure 4 shows the actual value and the predicted value. Both lines are overlapping and hardly can see the blue line. Does it mean the authors have a prefect prediction? There were equations in Table 5; however, we read not specific explanation of exactly what are these equations for.

Last, the authors should pay more attention to its English writing as there are many grammatic mistakes, in the abstract it was seldomly using “you” or “your response”, which is quite confusing to the readers.